# Simulating HIV transmission dynamics: An agent-based approach using NetLogo

**Sophia Nicolette C. Amasa**[ID]◐, **Trisha Mae P. Beleta**◐,
**Shemaiah L. Montilla**◐, **Orven E. Llantos**[ID]◐*

Department of Computer Science, College of Computer Studies, MSU-Iligan Institute of Technology, Iligan City, Philippines

◐ These authors contributed equally to this work.
* orven.llantos@g.msuiit.edu.ph

**Data availability statement:** All relevant data are available in a public GitHub repository at: https://github.com/trishamee/HIV-Simulation-with-NetLogo. The dataset includes all materials

## Abstract

Many studies have employed Agent-Based Modeling (ABM) to study the complex dynamics of HIV transmission. However, these studies often focus narrowly on specific subpopulations and limited parameters, restricting the potential of ABM to capture the intricate interrelationships between diverse subpopulations. This paper proposes an improved ABM to simulate HIV epidemic dynamics, exploring parameters such as sexual behaviors, drug use, condom usage, testing frequency, and treatment-seeking behavior. Calibrated with empirical data from the Philippines, the simulation closely aligns with national HIV infection trends from 2010 to 2018, achieving a Mean Absolute Error (MAE) of 3.5 and a Mean Squared Error (MSE) of 14.9. Findings indicate that extended commitment duration, consistent condom use, sexual inactivity, high testing frequencies, and strict adherence to treatment significantly lowers HIV transmission rate. The simulation results mirror trends observed in other studies, suggesting that the enhanced model provides reliable and expected outcomes. The results also illustrate the relationships between different factors, highlighting the model's comprehensive approach. Furthermore, the model effectively captures the trends within a 10-year period, predicting the cyclical rise and fall of new infections every 2 to 3 years, along with the overall decline in the percentage of new infections over time. This paper represents an initial step, prompting further efforts to enhance understanding and public health interventions.

## 1 Introduction

Human Immunodeficiency Virus (HIV) is a major public health issue worldwide. It weakens the immune system by targeting key cells, increasing vulnerability to various infections and illnesses [3]. According to recent estimates by the United Nations (U.N.), nearly 39 million individuals were living with HIV in 2022 [4]. Particular concern surrounds specific subpopulations including sex workers, clients, people who inject drugs (PWID), men who engage in male-male sexual activity (MSM), and individuals with diverse gender identities such as transgender women and men, as they account for 70% of global HIV infections. In the Philippines, the situation is equally alarming, with the HIV and AIDS Repository Project (HARP) reporting 109,282 confirmed cases of HIV as of December 2022 [2]. The reported cases

necessary to replicate the findings of this study, including model parameters, simulation outputs, and source code for the NetLogo model.

**Funding:** Mindanao State University-Iligan Institute of Technology.

**Competing interests:** The authors have declared that no competing interests exist.

primarily affected males (94%), with sexual contact among males (MSM) being the leading cause (82%), followed by male-female sexual encounters (14%) and needle sharing (2%).

Understanding how the disease's spread is influenced by immunity, population density, and human behavior [52] is the key to guiding targeted public health and social measures (PHSMs) to effectively reduce transmission and community risk [51]. However, HIV transmission involves a complex interplay between individual behaviors, community interactions, and environmental factors [20–26] and understanding these complex dynamics require innovative research methodologies. Some of these methodologies include mathematical modeling [11,12,19], computational cohort modeling [13], generalized estimating equations [10], log-linear modeling [9], and stochastic modeling [14,15]. Because conventional methods often overlook relationship dynamics and population diversity, Agent-Based Modeling (ABM) has emerged as a powerful tool. ABM uses a bottom-up approach to simulate individual agents and their interactions, offering detailed insights into HIV spread in diverse communities [7].

Other studies adopting ABM focus on specific subpopulations, such as people who inject drugs [17] and men who have sex with men [16,18], often concentrating on specific factors like testing patterns [18] or the effects of treatment involvement [16,17]. This narrow focus often limits ABM's ability to replicate the interactions between different subgroups and the interplay among various factors. This study addresses existing limitations by enhancing Wilensky's HIV transmission model in NetLogo—a widely-used platform for agent-based modeling that includes educational models such as one for HIV [7,29]. This original model has been adopted in prior research [5,6], but our work builds upon it by introducing three key factors: drug usage, gender dynamics, and treatment inclination. These modifications aim to simulate more realistic population interactions and intervention outcomes.

The improved model introduces drug usage as a new factor, including needle sharing as a transmission mode and involving the PWID (People Who Inject Drugs) community. Additionally, it incorporates gender dynamics by introducing three gender categories, adding depth to the population's coupling behavior, including both same-sex and opposite-sex sexual interactions. The model also addresses mitigation efforts by simulating treatment effects, aiming to provide a better understanding of how healthcare interventions can influence HIV transmission dynamics within the simulated population. These enhancements significantly impact the overall dynamics of the simulated community and help replicate the relationships between the different factors involved in HIV transmission. The insights gained from this endeavour could guide the development of targeted and effective public health interventions, making a valuable contribution to global efforts to mitigate this significant health challenge.

## 2 Model development

The proposed model buids upon Wilensky's foundational HIV simulation model developed in 1997 [29], which depicts individuals forming pairs based on coupling behavior to simulate sexual relationships. The former model included parameters such as condom usage and testing frequency. Building on this structure, the present study introduces several significant innovations to reflect more realistic transmission dynamics. One of the model's key improvements is the integration of gender dynamics through three distinct gender types. In this context, the term gender is used to represent sexual orientation—defined as an individual's enduring emotional, romantic, or sexual attraction to others—and not gender identity, which relates to one's internal sense of self [1]. While the terminology aligns with the model's original structure, this paper clearly distinguishes between these concepts and

focuses on sexual orientation as a behavioral factor in transmission. However, the implementation retains the variable name *gender*, adhering to its original usage before implementing modifications.

Additionally, a new gender type, person righty, is introduced to capture more diverse relationship dynamics, including individuals who engage with multiple gender groups. This extension enables the model to represent subpopulations such as bisexual men and MSM (men who have sex with men).The model also integrates intravenous drug use as a transmission pathway, introducing the PWID (People Who Inject Drugs) community into the simulation, enhancing the model's ability to simulate diverse routes of infection.

Lastly, the model accounts for individuals undergoing HIV treatment, recognizing the evolving landscape of healthcare interventions and their profound impact on the spread of the virus. This inclusion not only reflects the current state of HIV care but also considers the potential role of treatment in reducing the likelihood of transmission.

## 3 Netlogo in agent-based modeling

NetLogo is a multi-agent programming language and modelling environment developed at Northwestern University's Center for Connected Learning and Computer-Based Modeling [27]. It is frequently used for modelling, assessing, and testing ideas in social sciences, biology, ecology, economics, and urban planning [28]. The software has a vast library of pre-built models, these models illustrate the platform's versatility by covering various issues such as market behavior, disease propagation, and traffic flow [38].

NetLogo has become an essential tool in agent-based modelling (ABM) due to its accessibility and versatility, making it suitable for various disciplines.NetLogo's primary strength in ABM lies in its capacity to simulate complex systems of interacting agents. This capability is particularly useful for investigating emergent phenomena, where the interactions of many individuals lead to outcomes that are difficult to predict based solely on the characteristics of the individual agents [39].

### 3.1 Definition and attribute of individuals

In the simulation, each *person-shaped* entity represents an *Individual*. Each individual possesses a set of randomly generated attributes. Yet, the values are constrained using the `random-near` NetLogo function to ensure they remain close to the population's average. These attributes encompass gender, drug inclination, coupling inclination, commitment duration, condom-use tendency, and testing frequency.

The simulation classifies each individual into one of three genders. An individual assigned a gender value of 1 is considered female and represented with *person woman* shape. A gender value of 0 indicates the individual is male and is either assigned to one of the two shapes: person-righty or person-lefty. These two male types differ in sexual orientation behavior. . It is important to note that while these two shapes, *person righty* and *lefty*, may be visually challenging to distinguish, they serve as a practical means to depict couple graphics and ensure compatibility in hand positions. An individual with the *person righty* shape can couple with female and male individuals represented by the *person lefty* shape. Conversely, a male individual with the *person lefty* shape can only couple with male individuals depicted as *person righty*. In contrast, female individuals can exclusively form couples with male individuals having the *person righty* shape.

Furthermore, attributes like drug, coupling, and condom-use tendencies or inclinations reflect an individual's predisposition toward certain behaviours. Drug inclination determines

the likelihood of an individual engaging in intravenous drug use, coupling inclination influences the propensity to form sexual relationships, and condom-use inclination determines whether a couple will use condoms during sexual activities. Meanwhile, for HIV testing, the model accounts for testing frequency rather than individual tendencies, denoting how often an individual undergoes HIV tests annually. While the simulation initially sets most of these factors, other variables, such as the partner's HIV status and the appearance of symptoms, can further amplify these tendencies.

Apart from the attributes above, several variables are available to monitor an individual's status. For instance, the *drugs?* variable records whether the individual engages in drug use, with a value of True indicating drug use and False indicating non-use. *infected?* variable indicates whether the individual is infected (True) or uninfected (False). *known?* variable reflects the individual's awareness of the infection, with a value of True indicating awareness; otherwise, the infection is deemed unknown. The *treated?* variable represents whether the individual is undergoing treatment, with a True value indicating treatment and a False value indicating no treatment. Finally, the *infection-length* variable monitors the duration of infection in days, which affects the manifestation of symptoms and other variables and factors.

Several variables are also available for coupling. For instance, the *coupled?* variable indicates whether an individual is in a sexual relationship, while the *couple-length* variable tracks the duration of an individual's current relationship. The *commitment* variable determines how long an individual tends to stay in a sexual relationship. In contrast, the *coupling-tendency* variable represents the likelihood of an individual to form a sexual relationship. The *condom-use* variable indicates the percentage chance that an individual uses protection. Finally, the *partner* variable represents the current partner of an individual.

## 3.2 Functions

The model includes several functions that assign values to different individual-level variables, such as `assign-commitment`, `assign-coupling-tendency`, `assign-condom- use`, `assign-test-frequency`, `assign-drug- tendency`, and `assign-treatment-tendency`, with average values determined by the users.

The `assign-color` function visually represents the presence of HIV infection in the population by assigning colours to each individual based on their health status. Five colours represent the health statuses and attributes: green for uninfected individuals, yellow for individuals using drugs while uninfected, blue for asymptomatic individuals, red for infected individuals aware of their infection, and orange for individuals undergoing treatment for infection.

The `move` function enables individuals to relocate within the model space by randomly rotating and then advancing by one unit, allowing them to search for a new partner in a different location.

The `couple` function handles the coupling behaviour of *person righty* individuals and selects a potential partner from the neighbouring individuals. A potential partner must meet certain conditions, such as not being coupled and having a shape that is *person lefty* or *person woman*. If the simulation identifies a potential partner, they can couple based on the potential partner's coupling-tendency value. If the random value is less than the value of coupling-tendency, the individuals become a couple. The individuals then change their background colour to grey to indicate coupling and stop moving.

The `uncouple` function handles the uncoupling behaviour of the individuals, with a condition for uncoupling based on the couple-length and commitment values. The simulation

executes the uncoupling process if the couple-length exceeds the commitment value of the individual or their partner. The individuals then reset their couple-length to zero, and their background color reverts to the original color. The partner resets to nobody, and both individuals mark themselves as uncoupled, free to move around the model space and find new partners.

The `infect` function manages the HIV transmission process among individuals within coupled scenarios or among PWID individuals, considering variables such as condom use, infection probability, and drug use. The probability of transmission may decrease depending on the infected partner's awareness and treatment status. If an infected partner is aware, a tendency towards condom-use increases by 1, and if treated, the infection chance decreases by 20%.

Depending on the infected partner's awareness and treatment status, the probability of transmission may decrease, with an increased tendency toward condom use or lowered risk of transmission within treated scenarios.

The `test` function simulates HIV infection testing. Individuals undergo testing based on the value of their *testing-frequency* parameter, which is set on a scale of 0 tests per year, 1 test per year, or 2 tests per year. Moreover, if individuals are aware of their infected partner or if symptoms start showing, then testing may occur regardless of their set *testing-frequency*,

The `treat` function simulates the treatment behaviour of the individuals. If the *treatment-tendency* is greater than 50% and the individual knows they are infected, the simulation sets the variable *treated?* to true. Those with *treated?* set to true have a 20% lower chance of being infected and are counted in the treated population, helping to track the number of individuals who underwent treatment. Moreover, if an individual's symptoms are showing, treatment may also occur.

### 3.3 Simulation

Algorithm 1 outlines the entire process. The algorithm starts by configuring the average values for each parameter which includes creating the simulated population (line 1). The simulation begins by evaluating two conditions: whether all individuals are infected and aware of their status (line 3) and whether all infected individuals have received treatment (line 6). If either condition is met, the simulation concludes. Otherwise, it proceeds to the next steps.

In the first section (lines 10, 13, 16, and 19), the simulation manages duration and establishes relationships. It increments the infection duration for infected individuals (line 10) and increases the relationship duration for couples(line 13). These variables are critical for subsequent condition checks. Individuals not in relationships are relocated to new positions (line 16). If an individual desires to form a relationship with a neighbouring individual, a new couple may form (line 19).

The latter half of the simulation (lines 22, 25, 28, and 31) , focuses on managing relationships and health. This section utilizes functions such as `uncouple`, `infect`, `test`, and `treat`. Couples whose relationship duration exceeds either partner's commitment length will uncouple (line 22). HIV transmission can occur both between couples and among drug users (line 25). Testing is conducted based on test frequency, symptoms, and partner awareness (line 28). Treatment is administered according to each individual's inclination or symptom-presence (line 31).

Following these procedures, the simulation increments its tick (line 36) and loops again based on the updated status of the simulated population.

**Algorithm 1. Simulation.**

```
 1: CREATE SIMULATED POPULATION USING CONFIGURED PARAMETER VALUES
 2: while true do
 3:    if all individuals are infected and aware then
 4:       stop the simulation
 5:    end if
 6:    if all infected individuals get treated then
 7:       stop the simulation
 8:    end if
 9:    for all individuals do
10:       if individual is infected then
11:          increase infection duration by one unit
12:       end if
13:       if individual is in a relationship then
14:          increase relationship duration by one unit
15:       end if
16:       if individual is not in a relationship then
17:          move the individual to a new location
18:       end if
19:       if an individual is currently single, shows a ten-
   dency to initiate a sexual relationship, and is either
   heterosexual or homosexual then
20:          form a relationship
21:       end if
22:       if relationship is longer than either individual in a
   couple then
23:          Uncouple
24:       end if
25:       if individual is in a relationship with an infected
   partner or is a person who injects drugs (PWID) then
26:          transmit infection based on chance and preventive
   measures
27:       end if
28:       if testing frequency is once or twice a year, the
   partner is aware of the infection, or symptoms are present
   then
29:          Perform testing per specified frequency, or in
   response to other conditions per chance
30:       end if
31:       if treatment inclination is greater then 50% or
   symptoms persists then
32:          undergo treatment
33:       end if
34:       UPDATE VISUAL: Adjust colors for each latest attributes
   and status
35:    end for
36:    PROGRESS SIMULATION BY ONE-TIME STEP.
37: end while
```

## 3.4 Model calibration

The calibration process includes adjusting the model's condition thresholds and incorporating parameter values based on Philippine data from different credible sources to guarantee its precision in reflecting real-world HIV trends. The process entailed comparing simulated outcomes with observed statistical data on HIV infection trends from the Philippines covering the period from 2010 to 2018 [40–48]. The aim is to calibrate these parameters to align closely with the real-world circumstances reflected in the HIV statistics of the Philippines. The changes to these parameters aim to closely replicate observed trends, ultimately improving the model's adherence to empirical data without extending into a validation process.

The researchers conducted 25 simulation runs using a population of 5000 individuals (see Fig 1) . Within this population, 8% were initialized as infected individuals [35], and 1.25% were initialized as drug users (estimate from [36,37]), closely reflecting real-life data from the Philippines. The default parameter values are in Table 1. It is important to note that these values are estimated, as the exact values are not easily accessible, and multiple studies show slightly varied statistical results.

## 3.5 Sensitivity analysis experimental setup

The study employed sensitivity analysis to investigate the potential effects of specific parameter values. Each test used different values to capture variations in different scenarios. To emphasize the impact of parameters, the researchers followed the One-factor-at-a-time (OFAT) approach. This deliberate method aided in isolating the influence of specific

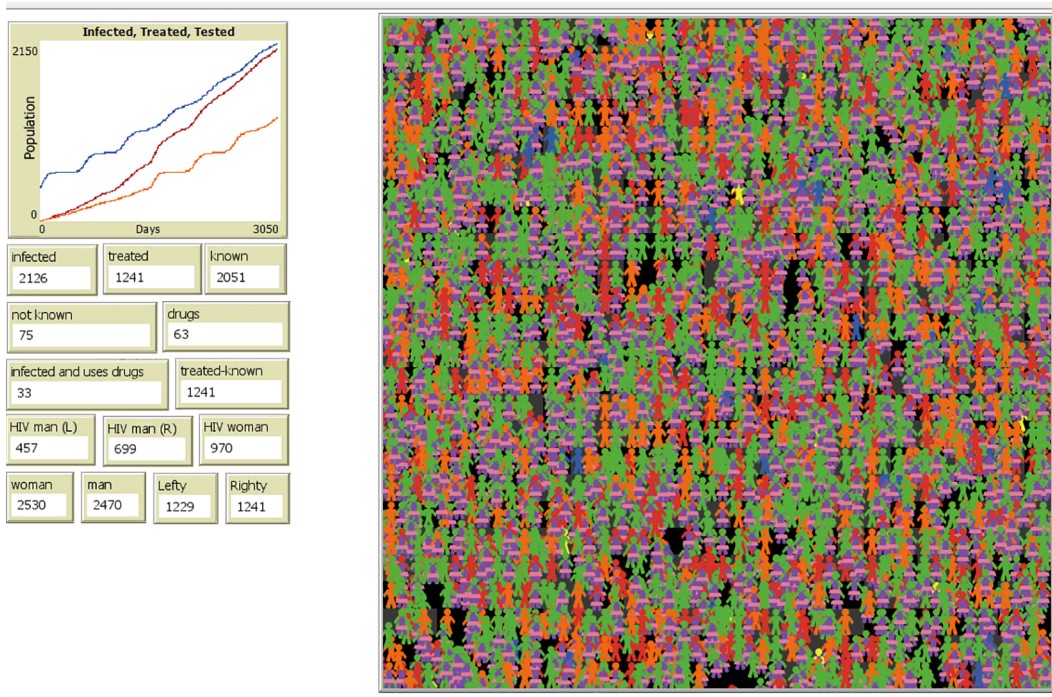

**Fig 1. Running simulation.**

**Table 1.** Default parameter values with sources.

| Parameters | Values | Sources |
|---|---|---|
| average-coupling-tendency | 30% | [30] |
| average-commitment | 523 days | [31] |
| average-condom-use | 40% | [34] |
| average-drug-tendency | 1% | [32] |
| average-treatment-tendency | 40% | [34] |
| average-test-frequency | 1 time(s)/year | [33] |

parameters on model outputs and mitigating potential confounding effects, thus ensuring a comprehensive understanding of their impact.

In this paper, varying values of each factor are referred to as *tendencies* or *inclinations*, representing variations in individual attitudes toward certain practices. These values are used as the average for the population in each scenario. Commitment length has six scenarios: average lengths of 0, 280, 560, 840, 1120, and 1400 days. In the simulation, each tick represents a day. Condom-use tendency also has six scenarios: 0, 2, 4, 6, 8, and 10, corresponding to 0%, 20%, 40%, 60%, 80%, and 100% inclination to use condoms. A scale of 1-10 is used to follow Wilensky's original values. Test frequency includes three scenarios: never, once a year, and twice a year. Treatment, coupling, and drug-use tendencies have six scenarios each: 0, 2, 4, 6, 8, and 10, following the same logic as condom-use tendency.

The researchers conducted a total of 825 simulated runs for the sensitivity analysis. Five parameters underwent six scenarios, while the test frequency had three. Each scenario was simulated for 25 runs using the default parameters in Table 1, with a population of 5000 with 8% initially infected. The percentage used is based on the HIV prevalence in the Philippines [35]. All parameters except the drug-tendency parameter had 1.25% initially drug-using individuals.

Statistics such as Pearson correlation and Spearman correlation is used to analyze the results. All computations are performed in Python using the `scipy.stats` module. For reference, the formulas used for these statistical measures are as follows:

The formula for the Pearson correlation coefficient is given by:

$$r = \frac{\sum_{i=1}^{n}(x_i - \bar{x})(y_i - \bar{y})}{\sqrt{\sum_{i=1}^{n}(x_i - \bar{x})^2 \sum_{i=1}^{n}(y_i - \bar{y})^2}} \tag{1}$$

where $x_i$ and $y_i$ are the individual sample points for variables $x$ and $y$, and $\bar{x}$ and $\bar{y}$ are the means of $x$ and $y$, respectively.

The formula for the Spearman rank correlation coefficient is given by:

$$\rho = 1 - \frac{6\sum_{i=1}^{n} d_i^2}{n(n^2 - 1)} \tag{2}$$

where $d_i$ represents the difference between the ranks of each pair of values, and $n$ is the number of observations.

## 4 Results and discussion

Upon comparing simulated outcomes with actual data, the researchers analyzed the annual percentage increase in the infected population, explicitly focusing on Philippine data from

2010 to 2018. The model indicated a Mean Absolute Error (MAE) of 3.5 and a Mean Squared Error (MSE) of 14.9.

The formula for Mean Absolute Error (MAE) is given by:

$$\text{MAE} = \frac{1}{n} \sum_{t=1}^{n} |A_v - P_v| \qquad (3)$$

While the formula for Mean Squared Error (MSE) is given by:

$$\text{MSE} = \frac{1}{n} \sum_{t=1}^{n} (A_v - P_v)^2 \qquad (4)$$

where $n$ is the number of years, $A_v$ is the actual value, and $P_v$ is simulated value.

The 3.5 MAE and 14.9 MSE can be attributed to the variability in data from various sources and the complexity of HIV transmission dynamics, which involves multiple interconnected variables. Nonetheless, the model effectively captures the general trends. Fig 2 illustrates these observations. Since the simulation population size and the actual population are not proportional, the yearly percent increase in infections was used for comparison. The mean number of infected individuals in the actual data was 17.74, with a standard error of 2.41 and a 95% confidence interval of 12.18–23.30. The simulated data had a mean of 18.12, a standard error of 2.54, and a 95% confidence interval of 12.27–23.97. The overlap in the confidence intervals suggests there is no statistically significant difference between the actual and simulated means.

In general, as the infected population grows, both the treated and tested populations also increase. However, the treated population follows a similar trend to the infected population,

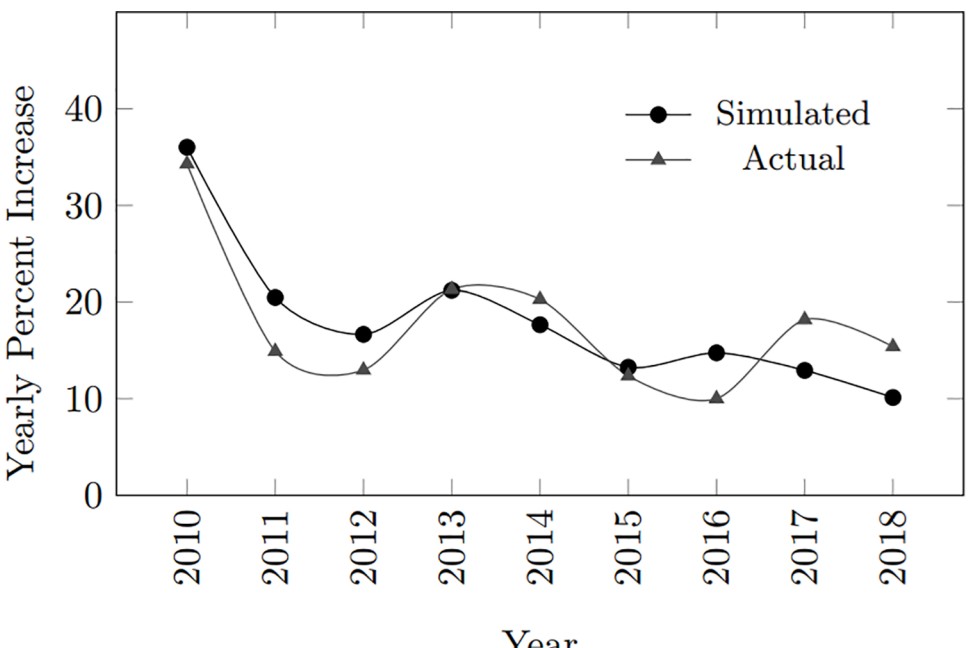

**Fig 2. Yearly percent increase in Philippine infected population from 2010 to 2018.**

while the tested population demonstrates a steeper and more rapid rise. The upward trend in treated individuals suggests a consistent improvement in treatment coverage within the simulated community, keeping pace with the increase in infected individuals. However, despite treatment, the persistent gap between the infected and treated populations indicates ongoing virus transmission, highlighting potential areas for improvement in preventive measures. Furthermore, the trend in tested individuals suggests that as the population of infected individuals expands, more individuals are inclined to undergo testing. Fig 3 depicts these findings.

Additionally, the count of drug users in the population remained constant throughout the simulation, attributed to a negligible average drug tendency among the population. Nevertheless, the escalation in infected drug users over time remains distinctly noticeable (see Table 2).

The trends observed in the simulation closely resemble those seen in actual Philippine statistics, although there are minor discrepancies. These differences could be due to the fixed

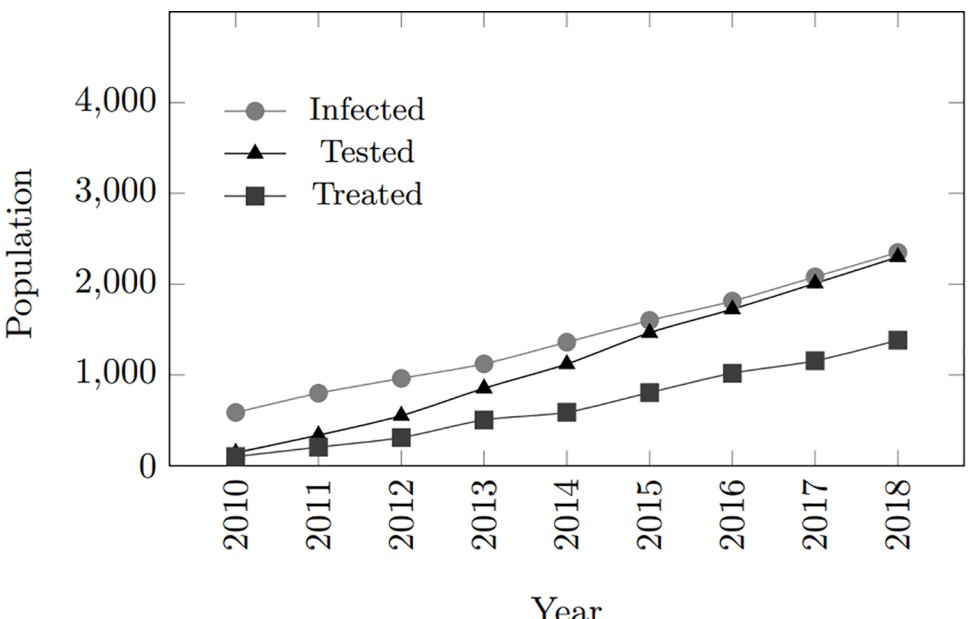

**Fig 3. Model calibration: Infected, tested, treated population within 10-year period.**

**Table 2**. Averaged simulation results.

| Year | Treated | Infected | Drug User | Tested | Infected Drug User |
|------|---------|----------|-----------|--------|--------------------|
| 1 | 101.36 | 587.28 | 63 | 144.36 | 12.12 |
| 2 | 205.6 | 798.8 | 63 | 337.44 | 14.44 |
| 3 | 310.76 | 962.32 | 63 | 552.2 | 16.84 |
| 4 | 505.12 | 1122.6 | 63 | 853.28 | 18.8 |
| 5 | 589.96 | 1360.72 | 63 | 1120.64 | 21.68 |
| 6 | 806.52 | 1600.88 | 63 | 1464.96 | 24.2 |
| 7 | 1019 | 1813.08 | 63 | 1724 | 26.24 |
| 8 | 1156.08 | 2080.4 | 63 | 2008.8 | 29.2 |
| 9 | 1382.88 | 2349.32 | 63 | 2298.44 | 32.28 |
| 10 | 1644.8 | 2587.2 | 63 | 2556.32 | 34.52 |

population in the simulation, which does not account for the introduction of new susceptible individuals who lack treatment or increased preventive measures. Additionally, variability in real-world data may contribute to these differences. However, the results remain promising despite these factors, indicating the model's potential utility with further enhancements.

## 4.1 Sensitivity analysis results

Significant effects are evident in factors such as commitment length, condom-use tendency, drug tendency, coupling tendency, and treatment tendency. These variables have emerged as crucial determinants significantly influencing the trajectory of HIV transmission dynamics within the simulation. For instance, varying commitment lengths showed a discernible impact on infection rates over time, while condom-use tendencies displayed a clear inverse relationship with infection rates. Similarly, an increase in the population's drug inclination resulted in a noticeable concentration of infected individuals among drug users, leading to a higher proportion of drug users within the infected population. Higher coupling tendencies also resulted in higher infection rates. Lastly, higher treatment inclinations corresponded to lower infected populations, demonstrating a significant impact of treatment tendency.

The following subsections provide a more detailed discussion of the outcomes for each parameter tested.

**4.1.1 Commitment length.** The trend of infected individuals is affected by the commitment length of individuals. Both the Spearman and Pearson correlation scores indicate a negative relationship between commitment length and the infection rate, with values of -0.66 and -0.53, respectively. This suggests that as commitment length increases, the number of infected individuals tends to decrease.

From the graphs, it can be observed that the commitment length of 0 days results in a lower infection rate than longer commitment lengths – 280 days and 560 days (as seen in Fig 4). Couples with 0 days of commitment have limited opportunities to transmit infections, explaining this unexpected observation, as the relationship terminates before significant transmission can occur.

However, a commitment length of 0 days presents a higher infection rate than the remaining commitment scenarios of 840, 1120, and 1400 days, as seen in Fig 5, since despite having limited time, the short 0-day commitment period still has more probability of encountering HIV-positive individuals during sexual encounters rather than those who stay in significantly more extended times.

However, the trend indicates a tendency to contract the infection even at significantly more extended times. This finding captures the complex behaviors of human beings, such as the tendency to engage with other individuals while in a committed relationship. One example that supports the idea is that of sociosexuality, wherein an individual engages in a sexual relationship with another outside of a committed relationship or without emotional attachments, like infidelity [49].

Moreover, consistent with previous observations, the trends in the populations of treated and tested individuals align with the increased number of infected individuals. However, an interesting trend emerges in the tested population, particularly in scenarios with longer commitment lengths. In these cases, the trend in the tested population converges more with that of the infected population, as depicted in Figs 6 and 7.

This observation shows that in longer-term relationships, partners of people living with HIV (PLWHIV) become aware and more conscious of their own HIV status more quickly when their partner tests positive. This increased awareness leads to more frequent HIV testing among these couples. In contrast, in shorter relationships, partners are less likely

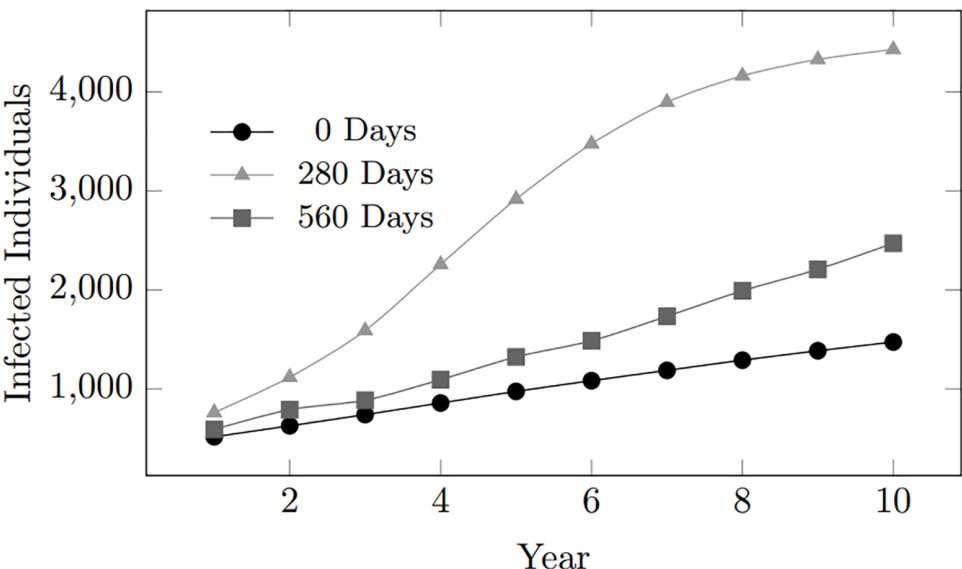

**Fig 4. HIV infections over 10 years by commitment length (0, 280, 560 days).** The simulation shows that relationship duration affects infection rates. Medium-term commitments (280 days) lead to the highest spread, while longer commitments (560 days) reduce infections. 0-day commitments show the lowest rates due to minimal repeated exposure.

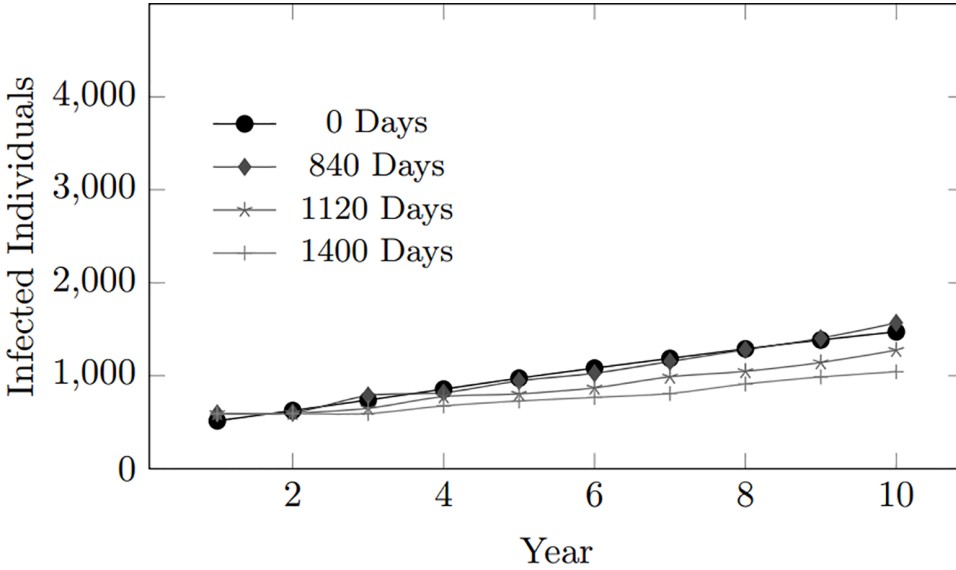

**Fig 5. Commitment length (0, 840, 1120, 1400 days): Infected population within 10-year period.** 0-day commitment shows higher infection rates due to increased chances of exposure across multiple partners as compared to much longer-term commitments.

to know about each other's HIV status, so their testing behavior depends solely on their personal preferences. All in all, the simulation results across these scenarios unequivocally illustrate a notable connection between commitment length and infection rates, highlighting the pivotal role of relationship duration in shaping the dynamics of HIV transmission.

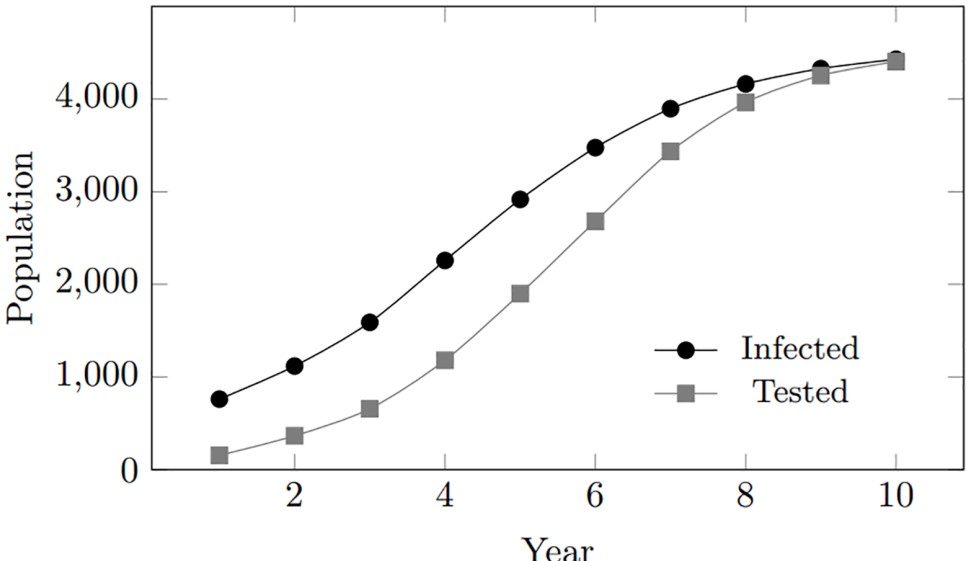

**Fig 6. Commitment length: Infected and tested population for commitment length 280 days.**

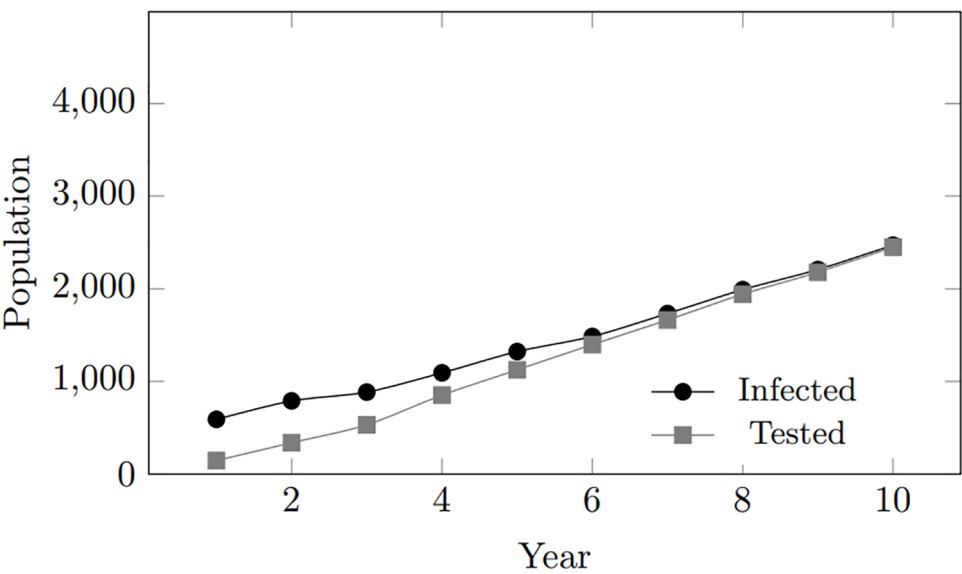

**Fig 7. Commitment length: Infected and tested population for commitment length 560 days.**

**4.1.2 Condom-use tendency.** The findings regarding the impact of condom-use tendencies on HIV transmission dynamics are unequivocal, demonstrating a clear inverse relationship between the inclination to use condoms and the infection rate. The condom-use factor exhibited a strong negative correlation with the number of infected individuals, achieving a perfect Spearman correlation score of -1.0000 and a high negative Pearson correlation score of -0.9037. This indicates that as condom use among couples increases, the number of new infections consistently decreases.

This pattern is evident in Fig 8 depicting the trend for the number of infected individuals within the population. Across different scenarios of varying condom-use tendency values

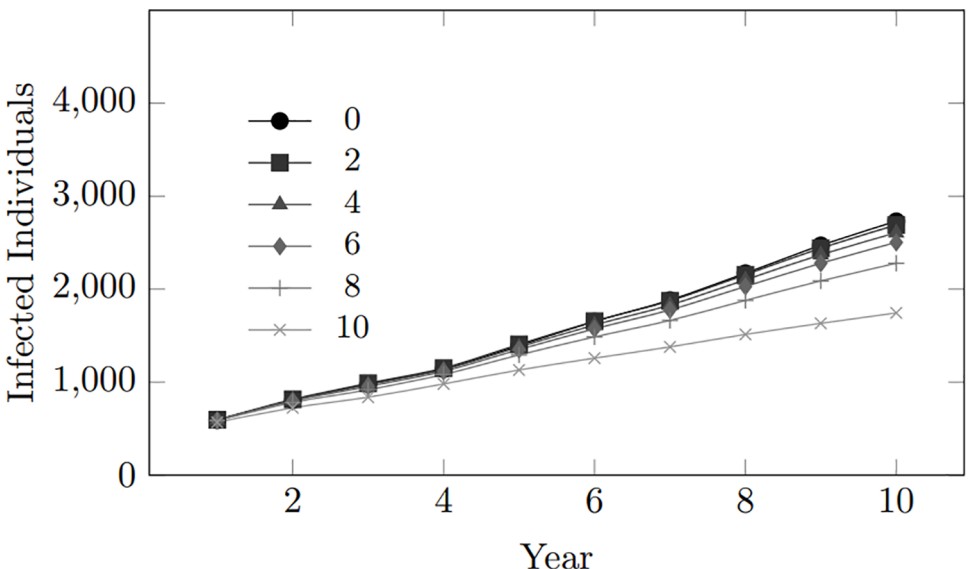

**Fig 8. Condom use tendency: Infected population within 10-year period.**

ranging from 0 to 8, there is a clear gradual difference in infection statistics, with higher condom-use tendencies consistently associated with lower infection rates. However, the most striking observation occurs at a condom-use tendency of 10 (depicted in Fig 8 by the line with the lowest trend), which depicts a condom-use inclination of 100%, meaning consistent use of condoms during coupling, where there is a significant reduction in the number of infected individuals. The findings emphasize the paramount significance of condom usage as an effective preventive measure against HIV transmission.

However, it is essential to note that high condom-use cannot guarantee the prevention of infection. Various factors, such as the effectiveness of the condom, still contribute to the spread of infection. Nevertheless, the result strongly supports the role of condoms as an effective preventive tool in mitigating the risk of HIV transmission. These findings highlight the importance of promoting consistent condom use as a key component of comprehensive HIV prevention strategies.

**4.1.3 Test frequency.** The study analyzed the simulation results for test frequency by comparing the percentages of aware and treated individuals within the infected population (see Table 3). In this approach, the aim is to discern trends within each scenario, particularly considering that individuals are marked as aware only if infected. The findings indicate that higher test frequencies increase awareness among the infected population. This observation shows that regular and consistent HIV testing helps people become more aware of their HIV status, which improves the overall effectiveness of HIV prevention and control efforts, such as an increased tendency to use condoms.

Table 3. **Testing frequency: Year 5 comparison.**

| Test Freq | Known vs Infected | Treated vs Infected |
|---|---|---|
| 0 | 82.13% | 42.85% |
| 1 | 82.34% | 42.90% |
| 2 | 82.40% | 43.04% |

However, the differences observed in awareness percentages are relatively small. This outcome can be attributed to the model's design, where an individual's inclination to testing is not the sole method for becoming aware of their HIV status. Factors such as partner awareness and the manifestation of symptoms also contribute to the awareness statistics. Additionally, the relatively small simulated population of 5000 individuals may contribute to challenges in clearly differentiating the impact of testing frequency behaviours under these conditions. The results show that when more individuals regularly engage in HIV testing based on their preferences, rather than waiting for symptoms or knowing a partner's status, the infection rate decreases. This highlights the importance of personal commitment to regular testing in reducing HIV spread.

**4.1.4 Treatment inclination.** Higher treatment inclinations are associated with lower infection rates and a larger treated population. This is evident from the correlation scores for treatment inclination, which show a strong negative correlation with the infected population—achieving a Pearson score of -0.900 and a Spearman score of -0.9429. Conversely, treatment inclination is positively correlated with the treated population, with a Pearson correlation of 0.8382 and a Spearman score of 0.6571.

As shown in Fig 9, higher levels of treatment inclination consistently correlate with decreased HIV infection rates over a decade. Notably, all scenarios demonstrate a clear pattern, wherein increased treatment inclinations correspond to reduced infection rates. This observation suggests a positive correlation between the level of commitment to treatment and the mitigation of new HIV infections.

Figs 10 and 11 provide a more precise depiction on the impact of high treatment inclination. These graphs reveal that elevated treatment tendencies correspond with a rise in treated individuals, indicating a proactive approach to managing HIV cases. In scenarios with higher treatment inclinations, individuals seek treatment promptly upon becoming aware of their infection. In simulations with a 100% treatment inclination, all individuals who test positive for HIV receive treatment promptly.

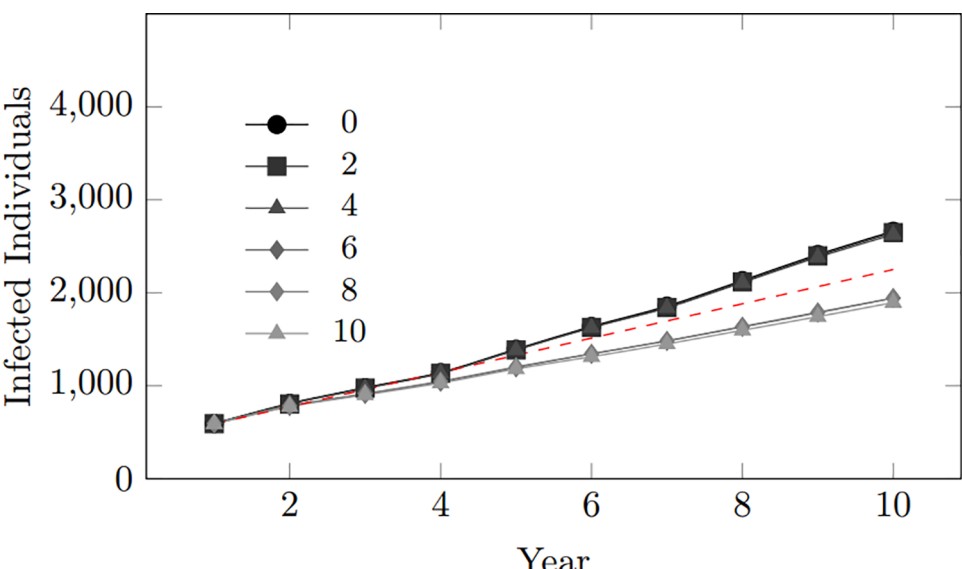

**Fig 9. Treatment inclination: infected population within 10-year period.** The dashed line separates scenarios with less than 50% treatment inclination (above the line).

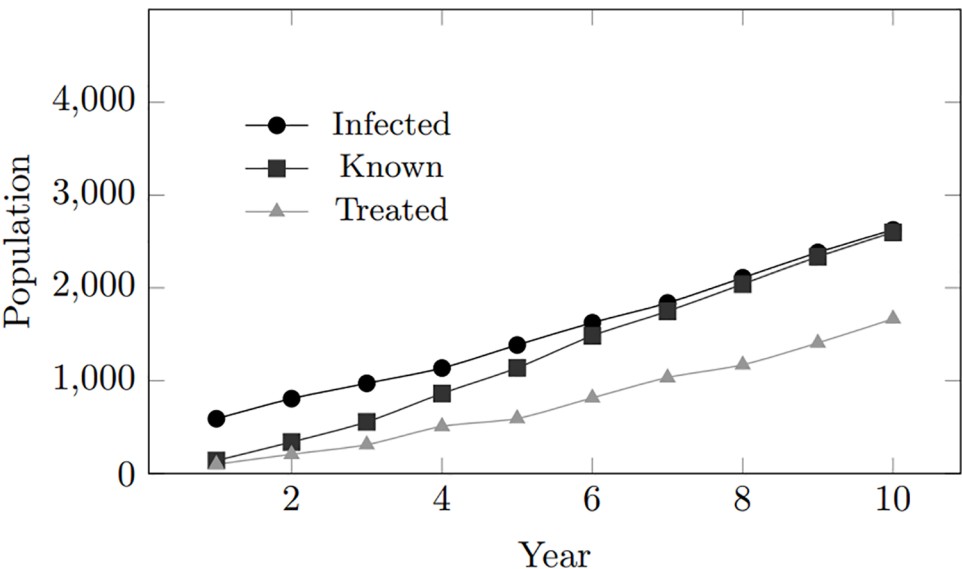

**Fig 10. Treatment inclination: Infected, treated, and tested population within 10-year period for treatment inclination 4.**

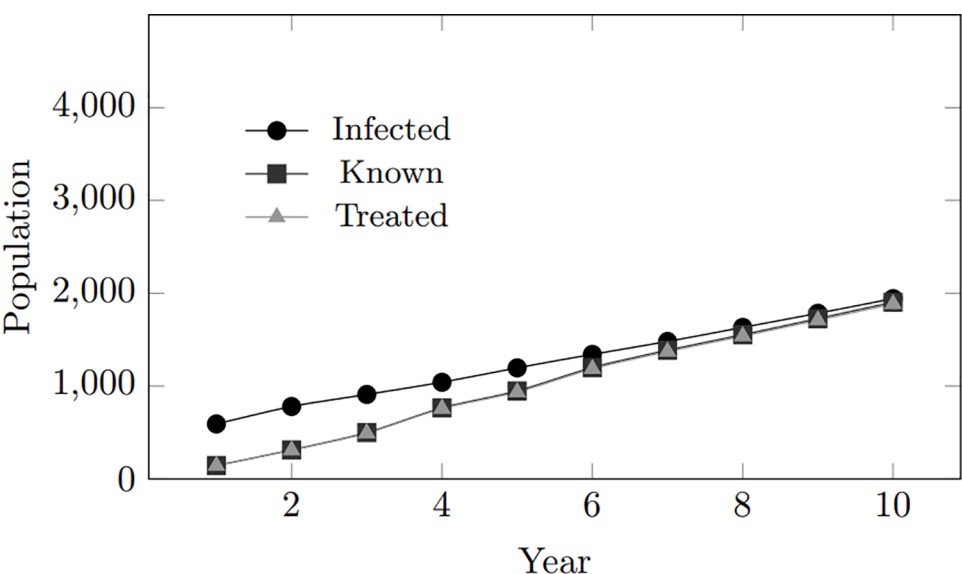

**Fig 11. Treatment inclination: Infected, treated, and tested population within 10-year period for treatment inclination 6.**

It is also important to emphasize that while treatment does not cure HIV itself, it effectively reduces the viral load, thereby decreasing the likelihood of transmission. These findings highlight the significance of treatment in shaping the dynamics of HIV transmission, associating higher treatment dedication with reduced infection rates.

Additionally, the relationship between treatment inclination and testing behavior is crucial. Effective mitigation of HIV depends not only on an individual's inclination to seek treatment but also on their testing behavior. Treatment only happens once an individual becomes

aware of their infection. When individuals get tested and start treatment upon discovering their infection, it significantly impacts their health and reduces the risk of transmitting the virus to others. In essence, the gap between the infected and treated population is influenced by the individual's willingness to seek treatment and their testing frequency.

**4.1.5 Drug tendency.**   Interestingly, infection trends do not vary significantly with different drug use tendencies, as shown by a low Spearman correlation score of 0.2. However, higher drug use tendencies are strongly associated with an increase in the number of infected individuals among People Who Inject Drugs (PWID), with a Spearman score of 0.9856 and a Pearson score of 0.8866.

This observation is evident in Fig 12 which reveals a significant rise in the number of infected individuals among People Who Inject Drugs (PWID). This suggests that the current infected population is predominantly comprised of PWID members, as shown in Fig 13.

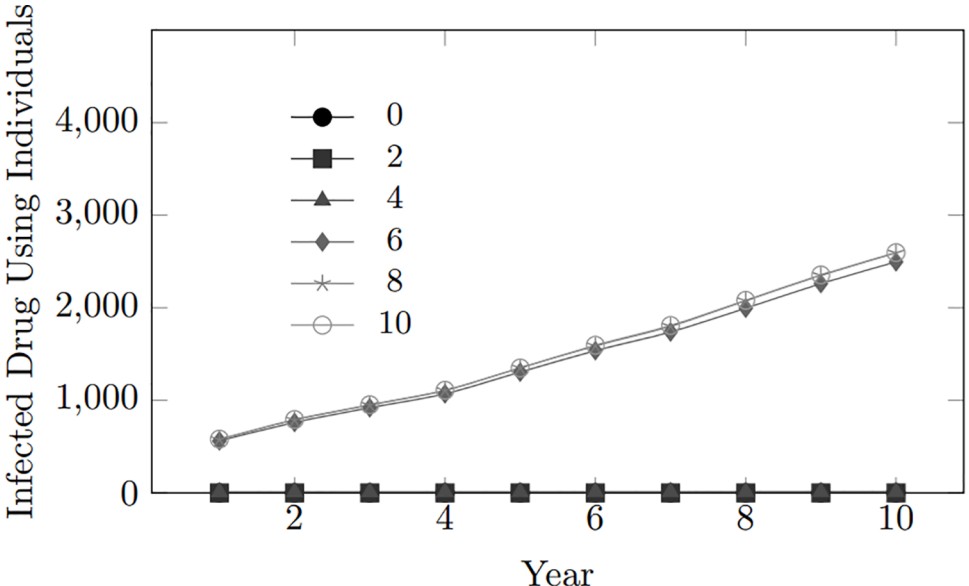

**Fig 12. Drug tendency: Infected drug user population within 10-year period.**

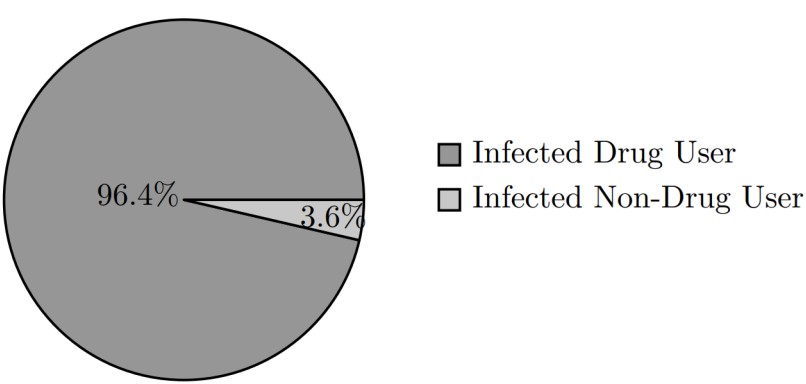

**Fig 13. Proportion of infected population that are drug users for drug tendency 6 year 10.**

Moreover, in these scenarios, values 0 to 4 remain flat, indicating that these lower inclinations are insufficient to prompt drug use.

Similar to earlier findings that emphasize the effectiveness of condom use and treatment in reducing HIV transmission rates, these results apply specifically to controlling transmission among couples within the subpopulation of People Who Inject Drugs (PWID). In a fixed population of drug users, the absence of new susceptible individuals means that preventive measures become more prevalent when a partner is infected. If a PWID partner becomes infected, preventive measures within the partnership intensify, effectively preventing an increase in infected individuals. Additionally, another factor contributing to the rising infection rates among populations with high drug tendencies is needle sharing among PWID.

The dual impact underscores the complex interplay of risk factors and preventive measures, with a potential scenario where one mode of transmission increases, but another is successfully controlled. The overall infection trends may not clearly show the effectiveness of preventive measures. Their impact might vary depending on the specific routes of transmission and risk behaviours within different subpopulations. The unique dynamics of drug injection practices, potential needle-sharing, and associated risk behaviours might render conventional preventive measures less effective in this context. It underscores the need for targeted interventions and tailored strategies to address the specific challenges posed by PWID populations in the context of HIV transmission.

**4.1.6 Coupling tendency.** Variations in coupling tendencies result in nearly uniform infection trends, as depicted in Fig 14, with values of 6, 8, and 10 showing a slightly higher infection rate. However, a notable difference is observed at a coupling tendency value of 0. In this scenario, where there is no coupling tendency—equivalent to sexual inactivity—the transmission dynamics are significantly altered. The infection trend is notably low, leading to the simulation ending prematurely after the third year due to the lack of new infections.

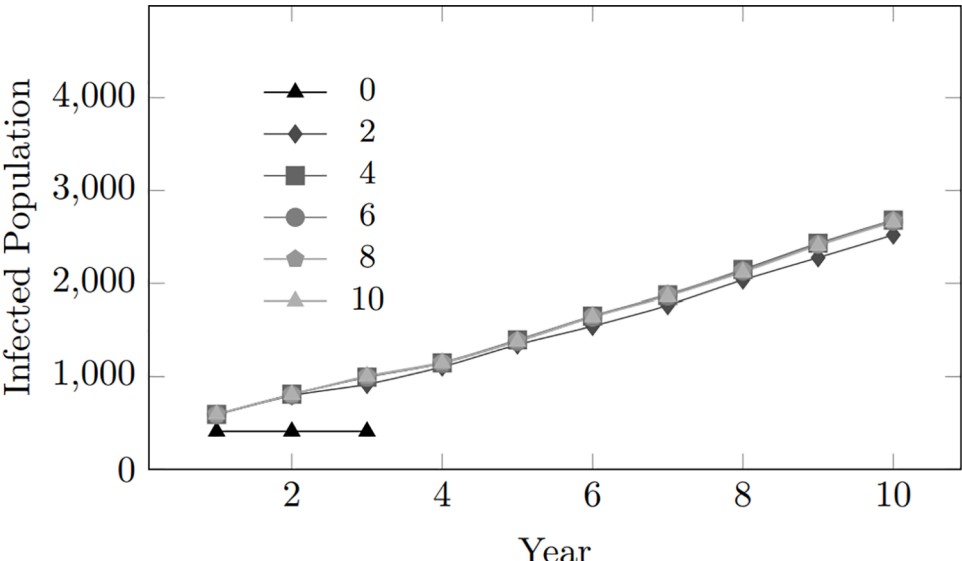

**Fig 14. Coupling tendency: Infected population within 10-year period.** Infection trends remain nearly uniform across coupling tendency values from 2 to 10, with slightly higher rates at 6,8,10. A clear deviation is seen at 0 coupling tendency, where the absence of pairing resulted in early simulation termination due to the lack of new transmissions.

Despite the overall uniform appearance of the graph for higher coupling tendencies, there is a strong positive correlation with the infected population. This is supported by Spearman's score of 0.9429 and Pearson's score of 0.7300, indicating that as coupling tendency increases, the number of infected individuals also rises. Additionally, a similar positive correlation is observed between coupling tendency and both the treated and tested populations, as detailed in Table 4. This underscores the significant impact that coupling tendency has on infection rates, treatment, and testing outcomes.

The exploration of coupling tendencies, which inherently involves relationship dynamics, is combined with an analysis of gender distribution within the infected population. Examining the distribution graphs below (Figs 15, 16), it is evident that despite variations in coupling tendencies, the simulation maintains a relatively equal distribution between genders in the infected population, given that the initial population consists of 50% Female and 50% Male (lefty and righty).

However, it is notable that a discernible difference in trend emerges beyond the initialized distribution issue of the gender population. Specifically, righty males (who can couple with both person-woman and person-lefty individuals) exhibit a higher proportion of the infected population than lefty males (who can only couple with person-righty individuals), despite both groups having the same initial population size. This may be due to the behaviour of righty males, who engage in riskier behaviours such as coupling with both genders, thereby increasing their likelihood of infection.

Apart from this particular trend, which closely mirrors gender dynamics, the main observation contrasts with real-world statistics, which often show an unequal distribution of HIV infections between males and females. This trend suggests that the current gender implementation in the model may require further refinement to align more closely with real-world

Table 4. Coupling tendency correlation scores.

| Outcome | Spearman Correlation | Pearson Correlation |
|---|---|---|
| Infected | 0.9429 | 0.7300 |
| Treated | 0.7143 | 0.6655 |
| Tested | 0.9429 | 0.7133 |

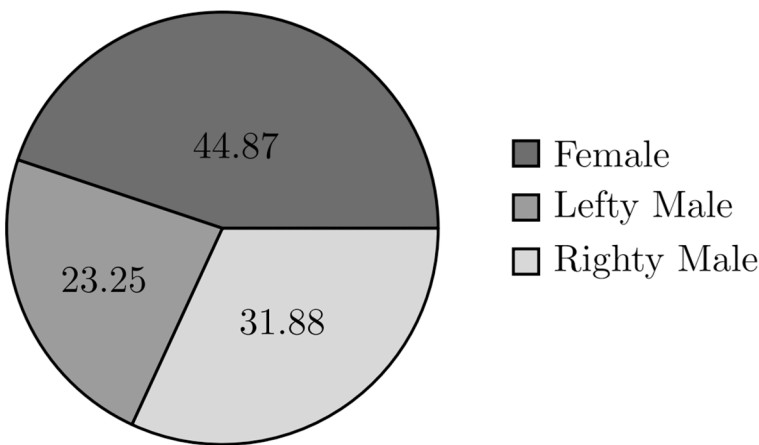

**Fig 15. Coupling tendency (8): Infected population per gender within 10-year period.**

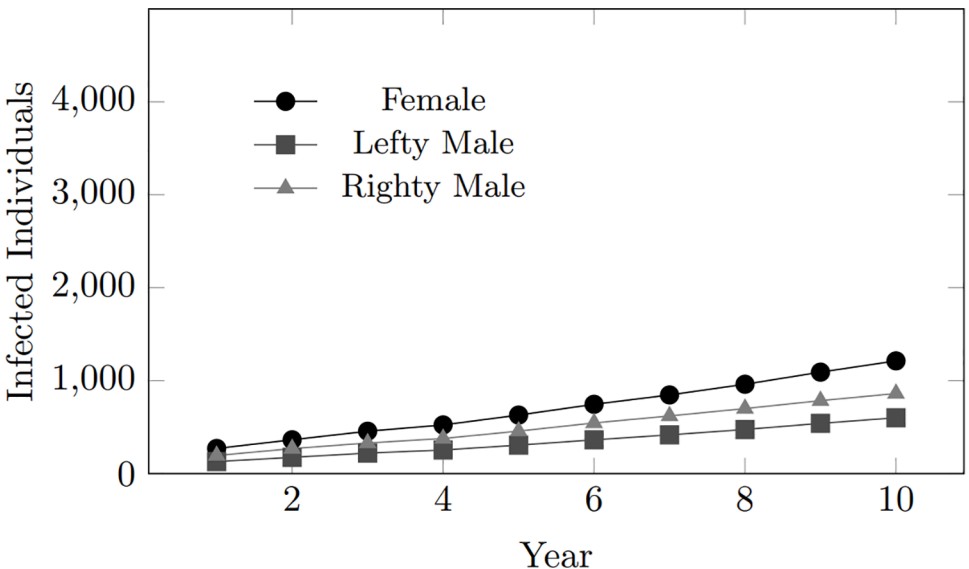

**Fig 16. Coupling tendency (8): Infected population per gender within 10-year period.**

scenarios. While the simulation captures relationship dynamics, it may not fully mirror the gender-specific patterns observed in actual HIV infection statistics.

## 4.2 Correlation results between various variables for number of infected individuals

The correlation results reveal that people who use condoms more frequently also tend to get tested more often and are more likely to seek treatment (see Table 5). Conversely, condom use decreases with higher levels of coupling, commitment, and drug use. Essentially, condom use behaviour aligns with other preventive health measures, such as testing and treatment, but diminishes as individuals engage in more sexual relationships. The strong correlations

**Table 5. Pearson and spearman correlations between various variables for the number of infected individuals.**

| Variable Pair | Pearson Correlation | Spearman Correlation |
|---|---|---|
| Coupling Tendency - Commitment | 0.9999 | 0.6957 |
| Coupling Tendency - Condom Use | -0.3168 | -0.7941 |
| Coupling Tendency - Test Frequency | -0.4472 | -0.7084 |
| Coupling Tendency - Drug Tendency | -0.0424 | -0.3189 |
| Coupling Tendency - Treatment Tendency | -0.3511 | -0.7941 |
| Commitment - Condom Use | -0.3247 | -0.7827 |
| Commitment - Test Frequency | -0.4519 | -0.5768 |
| Commitment - Drug Tendency | -0.0429 | 0.1429 |
| Commitment - Treatment Tendency | -0.3524 | -0.7827 |
| Condom Use - Test Frequency | 0.5220 | 0.7392 |
| Condom Use - Drug Tendency | -0.5425 | -0.1160 |
| Condom Use - Treatment Tendency | 0.7109 | 1.0000 |
| Test Frequency - Drug Tendency | 0.3460 | 0.3947 |
| Test Frequency - Treatment Tendency | 0.7796 | 0.7392 |
| Drug Tendency - Treatment Tendency | -0.2350 | -0.1160 |

with test frequency and treatment tendency highlight a consistent pattern of protective health behaviours among regular condom users.

Individuals who frequently get tested for infections are also more likely to use condoms and seek treatment. In contrast, those involved in multiple relationships or more committed to a single partner tend to test less often. Additionally, there is a moderate positive correlation between test frequency and drug use, suggesting that individuals with higher drug use may also test more frequently. Overall, the test frequency is positively associated with other protective health behaviours, particularly condom use and treatment-seeking.

Drug use shows a complex and generally weak relationship with other behaviours related to infection rates. While higher drug use is associated with less frequent condom use and a lower likelihood of seeking treatment, drug users tend to get tested more often. However, these correlations are generally weak, indicating that drug use does not strongly predict other behaviours, though some moderate trends are present, especially with condom use and test frequency.

People who regularly use condoms and get tested frequently are much more likely to seek treatment. Conversely, treatment-seeking decreases with higher levels of coupling and commitment, indicating that those more involved in relationships are less likely to seek treatment. The relationship between drug use and treatment-seeking is weak, suggesting that drug use has little impact on the likelihood of seeking treatment. Overall, people who seek treatment tend to engage in other protective health behaviours, especially using condoms and getting tested.

## 4.3 Model prediction

The model replicates the actual HIV trends for a 10-year period, from 2010 to 2020, as depicted in Fig 17. During these years, the simulated outcome aligns closely with real-world

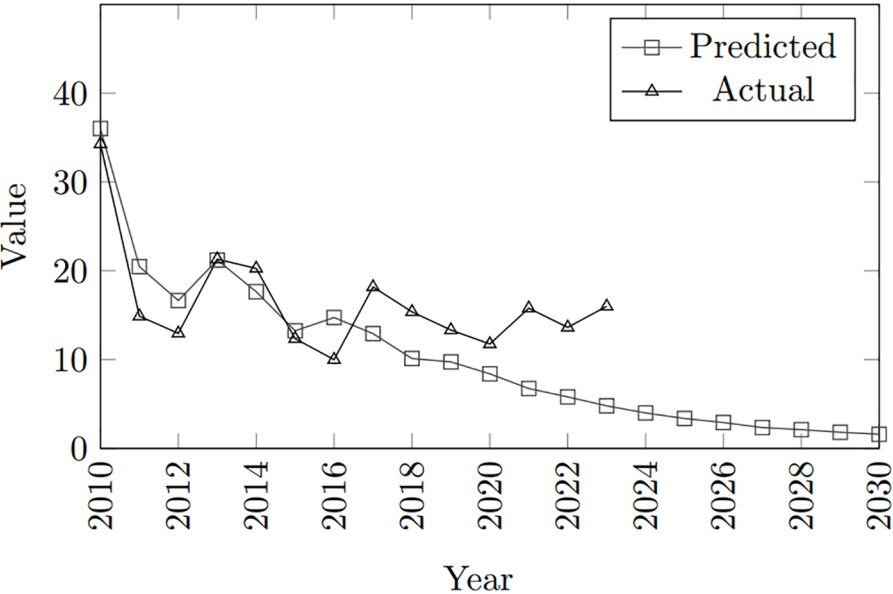

**Fig 17. Comparison of predicted and actual values of HIV transmission from 2010 to 2030.**

data, capturing infections' cyclical rise and fall every 2 to 3 years. This pattern reflects the dynamics of HIV transmission, where periodic fluctuations in new infections are observed.

Furthermore, although the magnitude of the peaks in the simulated data is less pronounced than in the actual data, the overall trend in both cases shows a consistent decrease in the percentage increase of new infections over time. For instance, the simulated model shows a percentage increase of 36.02% in 2010, which gradually declines to 8.04% by 2020. This is comparable to the actual trend, where the percentage increase decreases from 34.29% in 2010 to 11.76% in 2020. The model's ability to replicate this downward trend is a strong indicator of its effectiveness in capturing the essential dynamics of HIV transmission.

Additionally, the model demonstrates the gradual reduction in the magnitude of new infections, with each subsequent peak being lower than the previous one, reflecting the real-world impact of interventions and increasing awareness over time. This consistent alignment with the observed trend suggests that the model is capable in predicting short-term outcomes, making it a potentially useful tool for understanding the HIV dynamics in the Philippines.

While the model performs well over the initial 10-year period, capturing the cyclical rise and fall of infections, challenges arise when extending the simulation beyond this timeframe. The prediction of the infected population of HIV within a fixed population of 5,000 shows significant variation between the simulated and actual percentage increases over the years. In 2010, the simulated increase was 36.02%, while the actual increase was slightly lower at 34.29%. This trend continues, with the simulated and actual increases showing fluctuations over the years. Notably, the actual data shows a significant drop in the percentage increase from 2011 (14.89%) to 2012 (12.96%), while the simulated data presents a more gradual decline.

As the years progress, the gap between simulated and actual increases becomes more evident. For example, in 2014, the simulated increase was 17.65%, but the actual increase was higher at 20.27%. This divergence trend was further highlighted in 2017 when the actual increase spiked to 18.18%, contrasting with the simulated 12.93%. However, in 2021, the actual increase (15.79%) again surpasses the simulated increase (6.75%), indicating that the simulated model may not fully capture the factors driving the spread of HIV within the population.

By 2024, the simulated model predicts a much lower percentage increase (3.37%) compared to previous years, while the actual data shows a steadier trend, with an increase of 16% in 2023. The divergence between the actual and simulated data suggests that while the simulation provides a general downward trend in the percentage increase of HIV infections, it may not accurately predict the specific fluctuations and spikes observed in the data.

These observations indicate that the current model is effective for fitting trends over a decade but may not be suited for long-term projections beyond a 10-year period. The model's limitations are partly due to the small population size of 5000, chosen to accommodate computational constraints. This small population restricts the model's ability to accurately capture new infections among individuals who have not yet been exposed to preventive measures and treatments. As a result, while the model can reflect general trends, it may not fully represent the complex dynamics of HIV transmission in larger populations or over extended timeframes. Future work could benefit from incorporating larger populations and more dynamic factors to improve prediction accuracy for longer-term trends.

## 5 Conclusion

This paper attempted to create an improved agent-based simulation model in NetLogo, focusing on various subpopulations to analyze HIV transmission dynamics. It investigates the

influence of different factors on HIV spread within a simulated community, calibrated using real-world data from the Philippines. The model closely matches the country's HIV infection trend from 2010 to 2018, with a Mean Absolute Error (MAE) of 3.5 and a Mean Squared Error (MSE) of 14.9. Findings indicate that several factors, such as longer commitment duration, consistent condom usage, sexual inactivity, frequent testing, and strict adherence to treatment, can decelerate HIV transmission.

Additionally, an increase in the population of People Who Inject Drugs (PWID) contributes to a faster virus spread. However, preventive measures during sexual activity help mitigate the spread through the sexual mode of transmission. Other findings also reveal a link between gap in treatment coverage among the infected population and levels of HIV testing and awareness. Moreover, partners of people living with HIV (PLWHIV) who are in longer-term relationships tend to become more aware of their own HIV status faster when their partner tests positive. This increased awareness leads to more frequent testing and better adherence to preventive measures.

Moreover, the model prediction effectively replicates decade-long trends, capturing the cyclical rise and fall of infections every 2 to 3 years. Notably, the model also mirrors the observed reduction in the magnitude of new infections over time, indicating that while the infection rates still fluctuate, the peaks become progressively smaller. This outcome demonstrates the model's ability to simulate key patterns and trends, reinforcing its utility for short-term HIV transmission projections. Overall, the study clarifies the key variables affecting HIV transmission and provides valuable insights for developing effective prevention strategies and understanding the interrelationships between these factors.

It is important to note that although the simulation effectively explores HIV transmission dynamics, it has limitations in representing gender-specific trends in HIV infections. This implies that the model's current gender implementation may need further improvement to better correspond with real-world scenarios. This study also embraces several limitations, including the model's assumption of a constant 80% condom effectiveness [8], random allocation of individual parameters might overlook outliers, absence of testing preferences, and assumptions on the needle sharing behaviour among all drug users. Furthermore, the model's predictions are constrained by its inability to accurately forecast future trends beyond a decade, given the small population size and lack of new individuals introduced into the simulation.

## 6 Recommendation

Enhancing the model with a larger and more dynamic population could improve its long-term projections. Future studies could extend the model to investigate several new concepts. These include exploring scenarios where agents engage in multiple sexual partnerships, introducing variability in condom use patterns every week, examining situations where agents opt for coupling but are sexual inactive, simulating instances where agents forget to adhere to their treatment regimen for specific periods, and modelling the progression of HIV infection to AIDS, along with the potential mortality outcomes for affected agents.

## Author contributions

**Conceptualization:** Orven E. Llantos.

**Formal analysis:** Orven E. Llantos, Trisha Mae P. Beleta.

**Funding acquisition:** Orven E. Llantos.

**Investigation:** Shemaiah L. Montilla, Sophia Nicolette C. Amasa.

**Methodology:** Orven E. Llantos, Shemaiah L. Montilla, Sophia Nicolette C. Amasa.

**Project administration:** Orven E. Llantos.

**Resources:** Orven E. Llantos, Trisha Mae P. Beleta, Sophia Nicolette C. Amasa.

**Software:** Trisha Mae P. Beleta, Sophia Nicolette C. Amasa.

**Supervision:** Orven E. Llantos.

**Validation:** Shemaiah L. Montilla, Trisha Mae P. Beleta.

**Visualization:** Sophia Nicolette C. Amasa.

**Writing – original draft:** Shemaiah L. Montilla, Trisha Mae P. Beleta, Sophia Nicolette C. Amasa.

**Writing – review & editing:** Orven E. Llantos.

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
