## [Decision Letter · Decision Letter 0]

30 May 2025

PONE-D-24-55952Simulating HIV Transmission Dynamics: An Agent-Based Approach Using NetLogoPLOS ONE

Dear Dr. Llantos,

Thank you for submitting your manuscript to PLOS ONE. After careful consideration, we feel that it has merit but does not fully meet PLOS ONE’s publication criteria as it currently stands. Therefore, we invite you to submit a revised version of the manuscript that addresses the points raised during the review process.

We look forward to receiving your revised manuscript.

Kind regards,

José Antonio Ortega, Ph.D.

Academic Editor

PLOS ONE

Journal Requirements:

“Mindanao State University-Iligan Institute of Technology”

5. We note that your Data Availability Statement is currently as follows: 

“All relevant data are within the manuscript and its Supporting Information files.”

6.  Please ensure that you refer to Figure 1 in your text as, if accepted, production will need this reference to link the reader to the figure.

7. We note you have included a table to which you do not refer in the text of your manuscript. Please ensure that you refer to Tables 2, 3 and 5 in your text; if accepted, production will need this reference to link the reader to the Table.

**Additional Editor Comments:**

While two referees accepted to review the paper, only one submitted their recommendations. The editor has also inspected the manuscript corroborating the detailed opinion of the reviewer on the merit of the paper so that a second review is not judged necessary. There are minor specific comments that should be addressed. 

Reviewers' comments:

Reviewer's Responses to Questions

**Comments to the Author**

1. Is the manuscript technically sound, and do the data support the conclusions?

Reviewer #1: Yes

2. Has the statistical analysis been performed appropriately and rigorously? 

Reviewer #1: Yes

3. Have the authors made all data underlying the findings in their manuscript fully available?

Reviewer #1: Yes

4. Is the manuscript presented in an intelligible fashion and written in standard English?

Reviewer #1: Yes

5. Review Comments to the Author

Reviewer #1: This study presents a timely and technically sound approach to modeling HIV transmission through an enhanced Agent-Based Model (ABM) developed in NetLogo. The authors extend Wilensky’s foundational model by incorporating behavioral parameters such as sexual behavior dynamics, condom usage, drug use, testing frequency, and treatment inclination—contributing valuable insights particularly in the Philippine context.

The manuscript demonstrates strong technical soundness. The simulation logic is coherent, the agent attributes are clearly defined, and the algorithm is systematically presented. The enhancements—especially those involving diverse sexual behavior interactions and the inclusion of the PWID (People Who Inject Drugs) population—add complexity and realism to the model. These modifications allow for a more holistic view of the interrelationships that drive HIV transmission dynamics, especially among key subpopulations.

The statistical analysis is appropriate and rigorously applied. Model calibration using Mean Absolute Error (MAE = 3.5) and Mean Squared Error (MSE = 14.9) demonstrates a good fit between the simulated trends and actual HIV infection data from 2010 to 2018. The use of sensitivity analysis through a One-Factor-at-a-Time (OFAT) method is methodologically suitable, and the application of both Pearson and Spearman correlation coefficients effectively highlights the influence of various behavioral factors. While the analysis is sound, the addition of confidence intervals or variability ranges for the simulation outputs would improve robustness. Furthermore, a split-sample validation (e.g., using part of the data for calibration and the rest for testing) could further enhance confidence in the model’s predictive capability.

The authors affirm that all relevant data are fully available within the manuscript and Supporting Information files, which meets data availability requirements for simulation studies. However, to promote reproducibility, it is recommended that the authors provide public access to the NetLogo model file and simulation outputs via a platform such as GitHub or Zenodo, along with usage documentation.

The conclusions are generally well-supported by the simulation results. The findings—showing that higher condom use, frequent testing, longer commitment duration, and treatment adherence correlate with reduced HIV infections—are consistent with existing literature and epidemiological expectations. The model successfully mirrors real-world HIV trends in the Philippines across a 10-year window, capturing periodic fluctuations and a gradual decline in new infections. Importantly, the authors acknowledge the model’s limitations in long-term prediction due to assumptions like fixed population size and lack of new agent entry, which is appropriate. Nevertheless, some conclusions, such as the impact of sexual abstinence, should be interpreted more cautiously to avoid overgeneralization.

In terms of presentation, the manuscript is written in standard academic English and is generally intelligible. However, minor revisions in grammar and writing style would significantly improve clarity and readability. Sections 1 (Introduction) and 2 (Model Development) contain several repetitive and overly long sentences that could be streamlined for better comprehension. Additionally, the consistent and precise use of key terms such as “gender,” “sexual orientation,” “coupling tendency,” and “commitment” is essential to avoid confusion, particularly for interdisciplinary readers. Terminology like “person righty/lefty,” introduced in Section 2.3 and Table 2, should also be clearly defined and used consistently throughout the manuscript. While the figures and tables are informative and relevant, several—particularly Figures 4, 14, and 17—would benefit from more detailed captions to enhance standalone understanding. Standardizing Y-axis scales across related graphs is also recommended to facilitate easier visual comparison across simulation results.

There are no ethical concerns with the study, as it does not involve human participants or identifiable data. The manuscript properly declares “N/A” for ethics approval. The authors also thoughtfully discuss model assumptions and limitations, including the simplification of gender interactions, static condom efficacy, and universal needle-sharing behavior among drug users. Their suggested future directions—such as modeling multi-partner relationships, HIV progression to AIDS, and behavioral variability—are thoughtful and commendable.

In summary, this is a strong and innovative manuscript with relevant findings for public health modeling. I recommend minor revisions, particularly to improve language clarity (Sections 1–2), refine interpretation (Section 4.2.6), enhance figure captions (Figures 4, 14, 17), and ensure consistent terminology (Section 2.3, Table 2). With these enhancements, the manuscript will be a valuable contribution to the literature on computational modeling of HIV transmission.

6. PLOS authors have the option to publish the peer review history of their article (what does this mean?). If published, this will include your full peer review and any attached files.

Reviewer #1: No

---

## [Author Response · Author response to Decision Letter 1]

7 Jul 2025

José Antonio Ortega, Ph.D.

Academic Editor

PLOS ONE

July 5, 2025

Manuscript ID: PONE-D-24-55952

Manuscript Title: Simulating HIV Transmission Dynamics: An Agent-Based Approach Using NetLogo

Subject: Response to Reviewers.

Dear Dr. Ortega,

We sincerely thank you and the reviewer for the valuable feedback, which has strengthened our manuscript. Below we detail the actions taken to address the requested revisions:

1. We have named the files appropriately as:

o Response to Reviewers (this document)

o Revised Manuscript With Track Changes

o Manuscript (clean version)

2. We followed the PLOS One LaTeX template in revising and formatting the manuscript.

3. We have included the required funder statement regarding Mindanao State University-Iligan Institute of Technology:

4. We confirm that the code and data are available at:

https://medium.com/@tri.beleta/simulating-hiv-transmission-dynamics-an-agent-based-approach-using-netlogo-2eab9790e37e

5. Figure 1 is now clearly referred to in the text.

6. Tables 2, 3, and 5 are all explicitly referred to in the text to ensure clarity and consistency.

7. We corrected the authorship details in reference number 3 to fix the previous typographical error.

8. We complied with the addition of confidence intervals as requested. However, implementing a split-sample validation was challenging because the model uses constant parameter values throughout the simulated timeframe, meaning there is insufficient variation to meaningfully validate on separate samples.

9. All other minor revisions have been incorporated in sections 1 and 2. In addition, sections 1 and 2 have been edited to address grammatical and clarity concerns. Table 2 and Figures 4, 14, and 17 are now fully defined and accompanied by detailed captions for clarity.

We appreciate the opportunity to revise and resubmit, and we hope the manuscript now meets the journal’s requirements.

Respectfully,

Orven E. Llantos, PhD

on behalf of all co-authors

---

## [Editor Report · Decision Letter 1]

1 Aug 2025

Simulating HIV Transmission Dynamics: An Agent-Based Approach Using NetLogo

PONE-D-24-55952R1

Dear Dr. Llantos,

We’re pleased to inform you that your manuscript has been judged scientifically suitable for publication and will be formally accepted for publication once it meets all outstanding technical requirements.

Kind regards,

José Antonio Ortega, Ph.D.

Academic Editor

PLOS ONE

Additional Editor Comments (optional):

The previous reviewer was not available. However, the editor has checked that the recommendations have been implemented meeting PLOS ONE requirements for publication.
---

## [Editor Report · Acceptance letter]

PONE-D-24-55952R1

PLOS ONE

Dear Dr. Llantos,

I'm pleased to inform you that your manuscript has been deemed suitable for publication in PLOS ONE. Congratulations! Your manuscript is now being handed over to our production team.

Kind regards,

on behalf of

Dr. José Antonio Ortega

Academic Editor

PLOS ONE